# The effect of spironolactone on cardiac and renal fibrosis following myocardial infarction in established hypertension in the transgenic Cyp1a1Ren2 rat

**C. J. Leader, G. T. Wilkins, R. J. Walker** *

Department of Medicine, University of Otago, Dunedin, New Zealand

* rob.walker@otago.ac.nz

**Data Availability Statement:** All relevant data are within the paper and its Supporting Information files.

## Abstract

### Aims

The renin-angiotensin-aldosterone axis plays a key role in mediating cardiac and kidney injury. Mineralocorticoid receptor antagonism has beneficial effects on cardiac dysfunction, but effects are less well quantified in the cardiorenal syndrome. This study investigated cardiac and kidney pathophysiology following permanent surgical ligation to induce myocardial infarction (MI) in hypertensive animals with or without mineralocorticoid receptor antagonism.

### Methods

Hypertension was induced in adult male Cyp1a1Ren2 rats. Hypertensive animals underwent MI surgery (n = 6), and were then treated daily with spironolactone for 28 days with serial systolic blood pressure measurements, echocardiograms and collection of urine and serum biochemical data. They were compared to hypertensive animals (n = 4), hypertensive animals treated with spironolactone (n = 4), and hypertensive plus MI without spironolactone (n = 6). Cardiac and kidney tissue was examined for histological and immunohistochemical analysis.

### Results

MI superimposed on hypertension resulted in an increase in interstitial cardiac fibrosis (p<0.001), renal cortical interstitial fibrosis (p<0.01) and glomerulosclerosis (p<0.01). Increased fibrosis was accompanied by myofibroblast and macrophage infiltration in the heart and the kidney. Spironolactone post-MI, diminished the progressive fibrosis (p<0.001) and inflammation (myofibroblasts (p<0.05); macrophages (p<0.01)) in both the heart and the kidney, despite persistently elevated SBP (182±19 mmHg). Despite the reduction in inflammation and fibrosis, spironolactone did not modify ejection fraction, proteinuria, or renal function when compared to untreated animals post MI.

**Funding:** CL was a recipient of a Department of Medicine, University of Otago PhD scholarship. This project was funded with grants from the Otago Medical Research Foundation - Laurensen Award, Lotteries Health NZ,Maurice & Phylis Paykel Trust. The funders had no role in the study design, data collection and analysis, decision to publish or preparation of manuscript.

**Competing interests:** The authors have declared that no competing interests exist.

## Conclusion

This model of progressive cardiorenal dysfunction more closely replicates the clinical setting. Mineralocorticoid receptor blockade at a clinically relevant dose, blunted progression of cardiac and kidney fibrosis with reduction in cardiac and kidney inflammatory myofibroblast and macrophage infiltration. Further studies are underway to investigate the combined actions of angiotensin blockade with mineralocorticoid receptor blockade.

## Introduction

Hypertension is a leading cause of left ventricular hypertrophy and heart failure (HF) [1]. The hypertrophied ventricle is initially able to compensate for the increased after-load but this is associated with progressive myocardial remodelling, including accumulation of fibroblasts and collagen formation. In time, this leads to a reduction in left ventricular (LV) compliance, diastolic dysfunction, and subsequent systolic dysfunction resulting in left ventricular decompensation and finally, heart failure [2–5]. Hypertension is associated with atherosclerosis [6] further increasing the risk of myocardial infarction. Along with the rising rates of hypertension, the prevalence of heart failure (HF) is predicted to increase by 46% over the next 15 years [7]. Hypertension is also associated with the development of chronic kidney disease (hypertensive nephrosclerosis). Further, chronic kidney disease is also recognised as a major risk factor for cardiovascular disease [7].

The cardiorenal syndrome (CRS) encompasses a spectrum of disorders involving both the heart and kidneys, conditions in which either renal impairment occurs as a result of cardiac dysfunction, or cardiac structure and function are negatively affected by renal disorders [8,9]. This is thought to be the result of one or more factors including (but not limited to) the activation of the sympathetic nervous system (SNS), hemodynamic cross-talk between the failing heart and the response of the kidneys (and vice versa), the renin-angiotensin-aldosterone system (RAAS), and mediators of inflammation and fibrosis [10–13]. Renal dysfunction is often observed in heart failure patients [14,15], and contributes to an increased risk of death [16–18]. There is a complex inter-relationship between the heart and kidney, particularly in patients with heart failure but the exact pathophysiological mechanisms of this association remain unclearly defined.

Mineralocorticoid receptor antagonists (MRA), such as spironolactone, have been shown (in a variety of animal models and in clinical trials) to slow the rate of development of cardiac fibrosis and remodelling after cardiac injury [19–25]. These actions are thought to be both dependent on and independent of the actions of angiotensin II [24,26–28]. While a small number of studies have explored the potential protective effects of spironolactone in the kidney [29–32], no studies to our knowledge have examined the actions of mineralocorticoid receptor antagonists on progressive cardiac injury and kidney injury after myocardial infarction in an animal model of established hypertension.

The aim of this study was therefore, to investigate changes in cardiac and kidney functional parameters and associated cardiac and kidney interstitial fibrosis, following an induced myocardial infarction in animals with established progressive hypertension, and how the resultant pathophysiology may be modified by treatment with the mineralocorticoid receptor antagonist spironolactone.

## Results

All animals showed comparable weight gain (20±5%) throughout the experimental period. Of the animals that underwent surgery, only two animals (one from each of the myocardial infarction groups) died (both within two hours of surgery), giving a surgical survival rate of 83%.

Following establishment of hypertension (Day 0), animals had a SBP of 143±21 mmHg and after 28 days this had increased to 174±11 mmHg (Table 1). The addition of daily dosing with spironolactone to hypertensive alone animals resulted in no significant differences across all physiological parameters measured, when compared to untreated hypertensive animals (Table 1).

### Myocardial infarction

Sham surgery resulted in no significant measured physiological differences from the hypertensive control animals (SBP (179±8mmHg vs 174±11mmHg), EF% (77±2% vs 82±2%), EDV (0.31±0.04ml vs 0.37±0.02ml), ESV (0.07±0.01ml vs 0.07±0.01), Na/K urinary ratio (0.63±0.2 vs 0.55±0.03), protein/creatinine ratio (17.2±2.8 vs 15.5±3.9) and serum creatinine (107.2 ±9.1μmolL$^{-1}$ vs 95.8±11.4 μmolL$^{-1}$)). Myocardial infarctions (MI) consistently spanned the anterior and lateral cardiac segments and included a portion of the posterior segment of the left ventricle. Despite established hypertension, these animals also showed a sustained reduction in SBP ($p<0.05$), reduced ejection fraction (EF) (with associated increased end systolic volume) and an increase in serum creatinine reflecting reduced kidney function when compared to hypertensive animals (Table 1). In contrast, animals treated daily with spironolactone following a MI (H+SP+MI group) resulted in no significant difference in SBP when compared to hypertensive controls after 28 days (Table 1), despite reductions in EF similar to untreated animals with a MI. Of further interest, the size of the MI was similar between spironolactone treated and untreated groups (33±6% vs 42±6% respectively; Table 1).

### Myocardial and renal cortical interstitial fibrosis

After 28 days, hypertensive animals demonstrated increased interstitial myocardial fibrosis (1.3±0.5%) and renal cortical fibrosis (2.5±0.9%) when compared to normotensive reference

**Table 1. Physiological data from hypertensive animals (H), hypertensive animals dosed daily with spironolactone (H+SP), hypertensive animals with a MI (H+MI) and hypertensive animals with a MI dosed daily with spironolactone (H+MI+SP), after 28 days.**

| | | H (n = 4) | H+SP (n = 4) | H+MI (n = 5) | H+MI+SP (n = 5) |
|---|---|---|---|---|---|
| SBP (mmHg) | Baseline (day 0) | 143±21 [139–148] | 154±23 [146–163] | 160±28 [146–172] * | 149±19 [142–155] |
| | 4 weeks | 174±11 [171–178] | 185±23 [176–193] | 155±22 [148–162] *** ### | 182±19 [177–188] ^^^ |
| EF (%) | | 82±2 [79–85] | 83±2 [80–86] | 33±15 [14–51] *** ### | 36±11 [22–51] *** ### |
| EDV (ml) | | 0.37±0.02 [0.33–0.4] | 0.31±0.01 [0.29–0.33] | 0.5±0.1 [0.37–0.64] ## | 0.4±0.08 [0.3–0.5] |
| ESV (ml) | | 0.07±0.01 [0.05–0.08] | 0.05±0.01 [0.04–0.06] | 0.34±0.1 [0.19–0.49] *** ### | 0.25±0.1 [0.17–0.33] * ## |
| Size of MI (%) | | | | 42±6 [32–52] | 33±6 [26–40] |
| Urinary Protein:creatinine ratio (mg. mmol$^{-1}$) | | 15.5±3.9 [9.2–21.7] | 14.9±5.1 [6.7–23] | 15.4±0.9 [14.2–16.5] | 18.1±4.5 [10.9–25.3] |
| Urinary Na:K ratio | | 0.55±0.03 [0.51–0.59] | 0.52±0.05 [0.45–0.6] | 0.67±0.06 [0.6–0.75] ** ## | 0.52±0.04 [0.45–0.57] ^^ |
| Plasma Creatinine (umol.L$^{-1}$) | | 95.8±11.4 [77.7–113.9] | 77.2±14.8 [53.7–100.8] | 131±3.6 [82.3–179.7] # | 82.8±15.8 [57.7–107.9] ^ |

Significant difference from H indicated by * (* $p<0.05$, ** $p<0.01$, *** $p<0.001$).

Significant difference from H+SP indicated by # (## $p<0.01$, ### $p<0.001$).

Significant difference from MI indicated by ^ (^^ $p<0.01$, ^^^ $p<0.001$).

Systolic blood pressure (SBP, mmHg) following the two-week establishment (Day 0) is also shown. Values are shown as Mean ± standard deviation with 95% confidence intervals shown in parenthesis. EF- ejection fraction, EDV–end diastolic volume, ESV–end systolic volume, MI–myocardial infarction.

ranges for this model (Fig 1B). Daily spironolactone therapy (following establishment of hypertension) for 28 days resulted in no significant change in cardiac fibrosis when compared to untreated hypertensive animals (Fig 1A and 1B). In contrast, the addition of spironolactone significantly blunted the progression of renal cortical fibrosis (reduced from 2.5±0.9% in hypertensive animals to 1.5±0.8%, p<0.05; Fig 1B and 1C).

The addition of a MI to established hypertensive animals significantly increased interstitial cardiac fibrosis (2.2±0.5%; p<0.001) and renal cortical interstitial fibrosis (3.1±0.9%; p<0.01) when compared to hypertensive animals (Fig 1). Daily treatment with spironolactone following a MI in hypertensive animals resulted in a significant (p<0.001) reduction in the amount of cardiac fibrosis (away from infarct) and renal cortical fibrosis (when compared with the untreated infarcted animals). The extent of cardiac and renal fibrosis in this group was similar to that seen in the hypertensive spironolactone treated animals (Fig 1). There was a correlation between remote cardiac fibrosis and cortical renal fibrosis ($R^2$ = 0.31, p = 0.019).

Myofibroblast infiltration (assessed by αSMA), into the areas of interstitial fibrosis, was significantly higher in hypertensive control animals than in those treated with spironolactone, both in the heart (24±16 vs 45±24 myofibroblasts/mm$^2$ respectively, p<0.001) and the renal cortex (1.2±0.4 vs 2.1±0.2 myofibroblasts/mm$^2$, respectively p>0.05, Fig 1). Macrophage infiltration in the myocardial interstitium was similar in both H and H+SP animals (50±17 vs 43±12 macrophage/mm$^2$, respectively), but spironolactone significantly reduced macrophage infiltration in the renal cortex (177±108 vs 111±59 macrophage/mm$^2$, respectively p<0.001, Fig 1).

In comparison with hypertensive animals, animals with a MI superimposed on hypertension resulted in a significant increase in interstitial cardiac myofibroblasts (45±24 vs 86±43

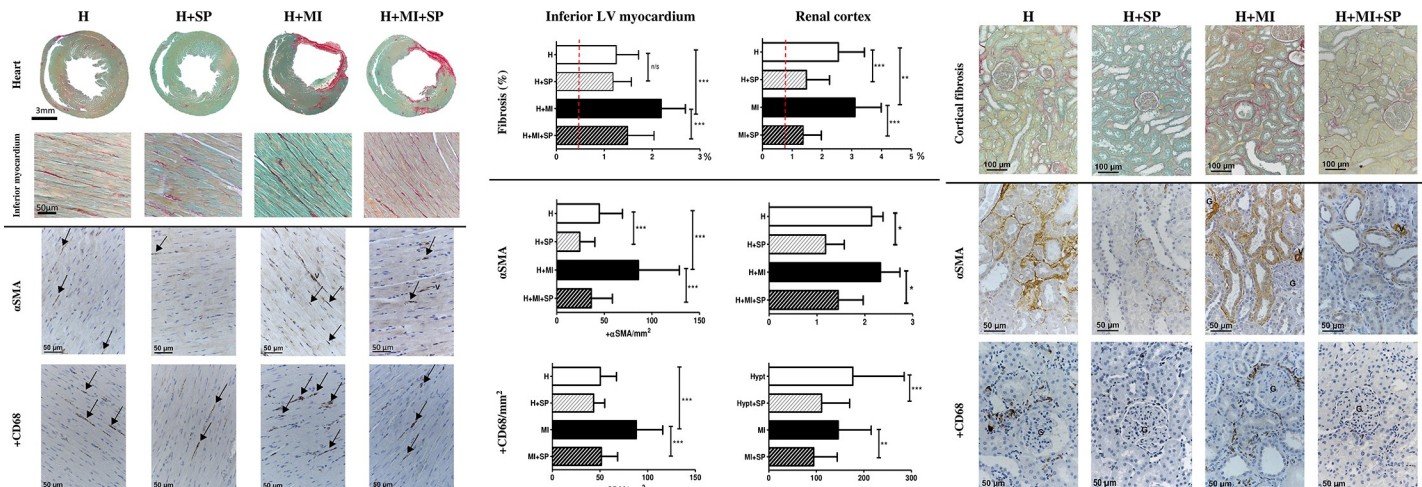

**Fig 1. Left ventricular remote cardiac and renal cortical interstitial fibrosis and associated myofibroblast and macrophage infiltration, from hypertensive animals (H), hypertensive animals treated with daily oral spironolactone (H+SP), hypertensive animals with a myocardial infarction (H+MI) and hypertensive animals with a myocardial infarction and daily oral treatment with spironolactone (H+MI+SP) after four weeks. (a)** Representative images of cardiac fibrosis, myofibroblasts and macrophage from the inferior segment of the left ventricle. The top panels show histological sections (Picrosirius red) of the whole heart (transverse section taken 6mm from the apex), with higher powered images (x100) of the inferior myocardium in the panels below. The lower panels show representative pictures (x100 magnification) taken from the remote inferior segment of myofibroblasts (stained positive with alpha smooth muscle actin; αSMA, and shown by the arrows) and macrophage (stained positive with Cluster of Differentiation 68; CD68, and shown by arrows). **(b)** Quantitative assessment of remote cardiac (left) and renal cortical (right) interstitial fibrosis (as assessed by Picrosirius red, top panels), myofibroblasts (as assessed by αSMA, middle panels) and macrophage infiltration (as assessed by CD68, bottom panels). The red dashed line represents the mean normotensive values (data not shown). Values are expressed as mean ± standard deviation. * indicates significance difference between groups (* p<0.05, ** p<0.01, *** p<0.001). **(c)** Representative images of renal cortical fibrosis, myofibroblasts and macrophage in the mid renal cortex. The top panels show renal cortical sections (x50 magnification) stained with picrosirius red. Middle panels show positive stain (brown) for myofibroblasts (αSMA; x100 magnification). Bottom panels show positive stain (brown) for macrophage (CD68; x100 magnification). V- vessel, G- glomeruli.

fibroblasts/mm$^2$, respectively. In contrast, this increase in myofibroblasts was not seen in the renal cortex (2.3±0.4 vs 2.1±0.2, respectively) after 28 days, despite a significant increase in total renal cortical fibrosis. A similar pattern for macrophage infiltration was seen in hypertensive animals with a MI, with a significant increase in cardiac fibrosis compared to hypertensive animals (88±28 vs 50±17 macrophage/mm$^2$, respectively, p<0.001), but no differences in macrophage infiltration in the renal cortex (146±69 vs 177±108 macrophage/mm$^2$, respectively) (Fig 1).

When compared to untreated animals with an MI after 28 days, administered spironolactone following a MI in the hypertensive animals resulted in a significant decrease in both the heart and the renal cortex of myofibroblasts (86±43 vs 36±22 fibroblasts/mm$^2$ (p<0.001) in the heart and 2.3±0.4 vs 1.4±0.5 fibroblasts/mm$^2$ (p<0.05) in the renal cortex) and macrophage infiltration (88±28 vs 51±17 macrophage/mm$^2$ (p<0.001) in the heart and 146±69 vs 94 ±50 macrophage/mm$^2$ (p<0.01)in the renal cortex) (Fig 1). In fact, these assessments were found to be similar to hypertensive animals treated with spironolactone that had not undergone an MI.

## Glomerulosclerosis

Glomerulosclerosis was evident in hypertensive animals (GSI of 1.2±0.06), which was significantly reduced by spironolactone (GSI of 0.89±0.04, p<0.01) (Fig 2). The addition of a myocardial infarct increased the extent of glomerulosclerosis (1.5±0.12, p<0.05) which was reduced by spironolactone treatment significantly (GSI 1.2±0.15 (p<0.01) (Fig 2). GSI was also found to be significantly correlated with renal cortical fibrosis (R$^2$ = 0.58, p = 0.0002), remote cardiac fibrosis (R$^2$ = 0.405, p = 0.006) and LVEF (R$^2$ = 0.36, p = 0.009) (Fig 2). Interestingly, despite the changes observed in glomerulosclerosis, there was no significant differences in urinary protein/creatinine ratio across the four groups.

## Discussion

In the present study, we used a rat model of inducible hypertension (Cyp1a1Ren2 rat) alongside the well-established surgical model of myocardial infarction (the ligation of the left anterior descending coronary artery) to replicate a more typical clinical scenario, to explore the pathophysiological interactions between hypertension and myocardial infarction on cardiac and kidney structure and function, as well as the effects of mineralocorticoid receptor antagonism.

Superimposing a MI on established hypertension, despite blunting the progressive rise in blood pressure seen in the hypertension alone animals, resulted in a significant increase in cardiac interstitial fibrosis, and renal cortical interstitial fibrosis and glomerulosclerosis, compared to that seen in hypertensive animals alone. This increase in fibrosis was also matched with a significant increase in myofibroblasts and macrophage infiltration in both the heart and the kidney. Kidney function measured by serum creatinine was also reduced. Of interest, despite the increase in glomerulosclerosis, proteinuria was not increased. This could be explained by the reduction in blood pressure that was observed in the MI animals.

Treatment with spironolactone following MI, significantly blunted the progression of fibrosis and inflammation (myofibroblasts and macrophages) in both the heart and the kidney, despite the SBP remaining elevated. In addition, while spironolactone resulted in no significant difference in ejection fraction, or proteinuria, there were significant improvements in serum creatinine when compared to untreated animals with a MI, although serum creatinine was not significantly different from the hypertensive control animals.

The LAD ligation surgery, resulting in a myocardial infarct in the left ventricle, is the most commonly used experimental model of heart failure or injury, and carries an accepted

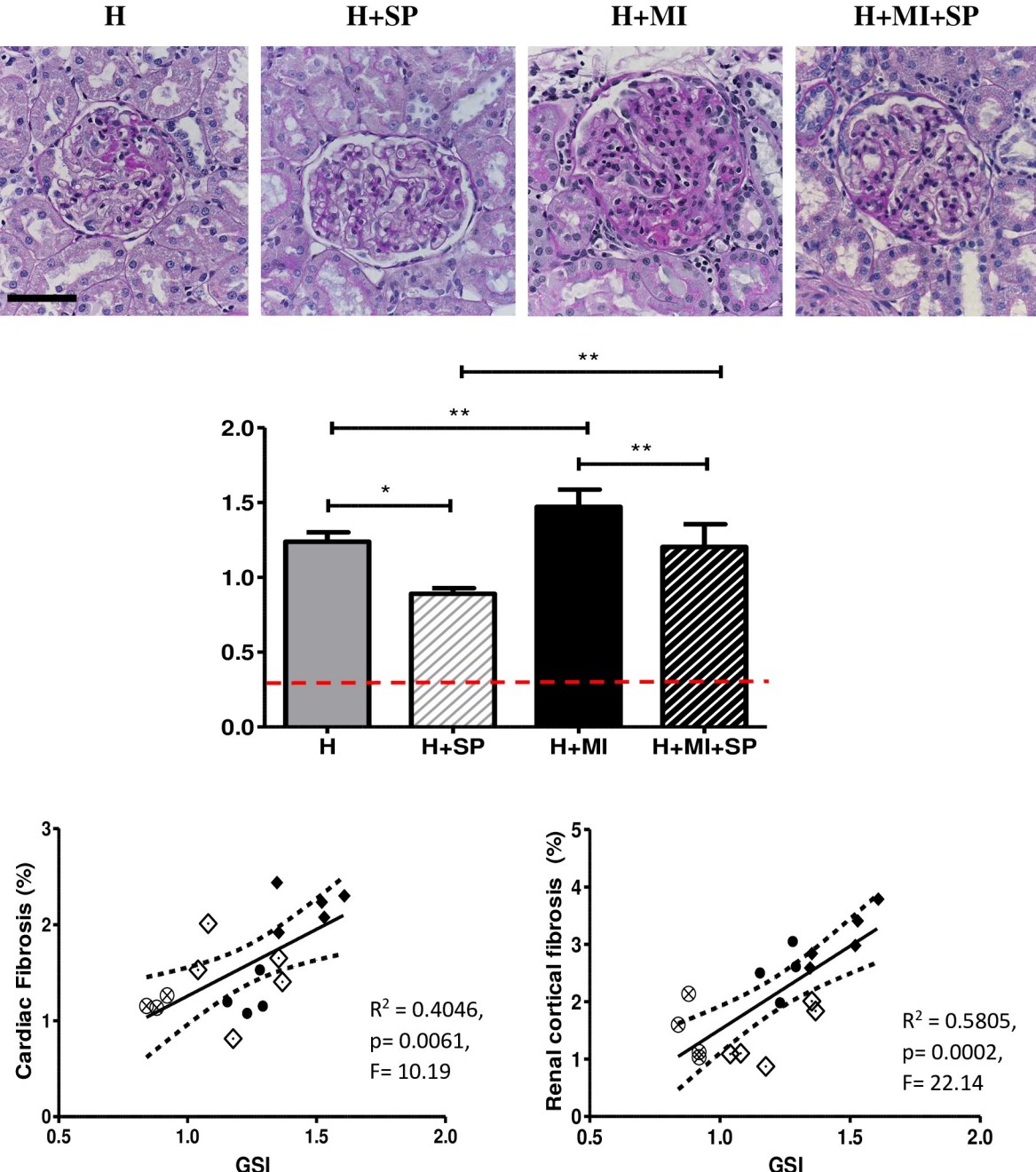

**Fig 2. Glomerulosclerosis in hypertensive animals (H), hypertensive animals treated with daily oral spironolactone (H+SP), hypertensive animals with a myocardial infarction (H+MI) and hypertensive animals with a myocardial infarction and daily oral treatment with spironolactone (H+MI+SP) after four weeks.** Top panels: Representative images of glomerulosclerosis (stained with PAS). Scale bar is 50μm. Middle panel shows the semi-quantitative assessment of sclerotic damage Values are expressed as mean ± standard deviation. Red dotted line is the mean Normotensive value (0.25±0.06; data not shown).* indicates significance difference between groups (* p<0.05, ** p<0.01, *** p<0.001). The bottom panels show the GSI correlation to cardiac interstitial fibrosis (left) and renal cortical interstitial fibrosis (right). H–Closed circles, H+SP–open circles with cross, H+MI–closed diamonds, H+MI+SP–open diamonds.

mortality rate of 50% [33]. Of note in our study, we had a mortality rate of just 17% despite the average infarct size being approximately 40%. It has been reported that the size of the infarct observed after 3–4 weeks can be considered a model of either myocardial infarction (infarct

size of around 20%) or heart failure (infarct size of 40–50%) [34–36], Myocardial infarction resulted in a significant increase in interstitial cardiac fibrosis, and renal cortical fibrosis and glomerulosclerosis, over and above that seen with hypertension alone, even despite the significantly reduced SBP. The current study showed no direct correlation between the overall size of the myocardial infarction to the degree of either myocardial fibrosis or renal cortical interstitial fibrosis. There was a significant correlation between the level of cardiac interstitial fibrosis and renal cortical interstitial fibrosis, although this relationship was not linear and is likely a highly complex relationship.

In many previously published studies, coronary ligation to induce myocardial infarction has been combined with a reduced renal mass model (by either uninephrectomy or sub-total nephrectomy) in order to mimic the cardiorenal syndrome [37–39]. Myocardial infarction superimposed upon sub-total nephrectomy, was associated with a greater degree of cardiac injury, more extensive glomerulosclerosis and renal interstitial fibrosis than that seen clinically [37–39] and often presented a greater degree of left ventricular dilation and progressive diastolic dysfunction, but not an increase in cardiac interstitial fibrosis or inflammation, when compared to animals with a MI only. In contrast, in our model with established hypertension, there was a greater degree of cardiac interstitial fibrosis and inflammation. The major difference relates in part, to the systolic blood pressure in these animals. The translational usefulness of these combined sub-total nephrectomy and MI models have been reviewed in depth by Bongartz and colleagues [40]. Bongartz and colleagues [39] showed that whilst their sub-total nephrectomy (SNx) animals generated hypertension (160 mmHg), in the combined SNx + MI group, systolic blood pressure fell below that of normal control animals, whereas in our hypertensive Cyp1a1Ren2 animals, systolic blood pressure even in the MI group remained elevated at 155±22 mmHg. Similar to our study, they reported a greater degree of glomerulosclerosis despite reduced blood pressure and cardiac output in the MI animals [39]. This was attributed to reduced nitric oxide availability and associated increased oxidative stress which exacerbated the cardiac remodelling and dysfunction along with the progressive glomerulosclerosis [39]. Although, markers of oxidative stress were not measured in our study, it is possible that the persistent hypertension in our animals further exacerbated the oxidative stress and subsequent cardiac and renal fibrosis that was evident.

The surgical interventions such as a reduction in kidney tissue or myocardial infarction are often performed on healthy organs where the compensatory regulatory mechanisms have not been activated [41] and do not accurately recapitulate the features seen clinically of chronic kidney disease. In the clinical setting however, the pathological insult (such as a MI) is usually preceded by a gradual loss of function of both the heart and kidneys usually in the setting of hypertension. By utilising an inducible hypertensive model and established hypertension prior to superimposing an acute myocardial infarction, this model may more accurately reflect the clinical situation.

Aldosterone has been demonstrated to mediate inflammation, tissue remodelling and fibrosis in many animal models of kidney and cardiac injury [26,42]. This may be due to upregulation of mineralocorticoid receptor expression in various vascular beds including the cardiac and renal microcirculation [43] as well as modulating macrophage function [44,45]. These effects can be both dependent and independent of angiotensin II [32,46–48]. The profibrotic effects of aldosterone are mediated in part via upregulating transforming growth factor β (TGFβ), and connective tissue growth factor (CTGF) expression promoting myofibroblast proliferation (αSMA expression) associated with increased collagen synthesis [49,50]. Mineralocorticoid receptor activation can also lead to increased oxidative stress inducing injury [51], as well as induction of podocyte injury leading to focal glomerulosclerosis [52].

Mineralocorticoid receptor antagonists (MRAs), have been demonstrated to attenuate or prevent the development of cardiac and kidney inflammation, fibrosis and failure, by

suppressing the over-activation of the mineralocorticoid receptor in multiple animal models [29,32,53–55]. Given the transgenic upregulation of renin and subsequently aldosterone in this animal model, we were keen to investigate the specific actions of mineralocorticoid receptor antagonism, utilising a clinically relevant dose of spironolactone in hypertensive animals subjected to myocardial infarction. As we have reported previously [56], most studies using rat models, have used spironolactone doses of between 20–200 mg/kg [29,48,53,57,58]. Using the allometric scaling calculations as described by Reagon-Shaw *et al* [59], a dose of 20-200mg/kg to a rat is a dose equivalent of 230-2270mg/day (respectively) for an average 70kg human. Spironolactone has poor water solubility but with good oral bioavailability with up to 90% absorption of spironolactone from the GI tract [60]. However, most animal studies utilise subcutaneous injections or osmotic pump for delivery, which makes translation to clinical practice more difficult. The oral dose used in this study was equivalent to a human dose of 50mg daily.

Treatment with a clinically relevant oral dose of spironolactone commenced following MI, significantly blunted the progression of fibrosis and inflammation (myofibroblasts and macrophages) in both the heart and the kidney, despite the persistently elevated SBP. Despite the reduction in inflammation and fibrosis (Figs 2 and 3), spironolactone did not modify ejection fraction, proteinuria, or renal function as measured by serum creatinine (Table 1) when compared to untreated animals with a MI.

There are some limitations related to this study. The number of animals varied from 4 to 8 in the different groups, however, given that they are a highly inbred strain, individual genetic variation is likely to be minimal. Significant results were obtained with these numbers. Also similar studies, using this model, have published significant data using only 3 to 5 animals in the different experimental groups [32,46]. Animals were only followed out to 4 weeks following myocardial infarction and longer-term follow-up in these animals would also be beneficial.

Whilst spironolactone reduced infarct size, there was smaller improvements in cardiac function. This may in part be explained by the limited echocardiographic views possible with a small animal. In addition, the damaged anterior wall of the myocardium became adherent to the rib cage which would also impact upon cardiac function. This would also limit the interpretation of the functional studies. Likewise, studies with combined angiotensin receptor blocking agents and mineralocorticoid receptor antagonists would further our understanding of the pathophysiology of this cardiorenal model.

While data from animal studies have shown that spironolactone reduces cardiac fibrosis and improves cardiac dynamics [24,25], albeit with rather high translational doses [61–64]. Large clinical studies such as RALES [65] and EPHESUS [66] have demonstrated the efficacy of mineralocorticoid receptor antagonism in patients with heart failure showing that low dose mineralocorticoid receptor antagonism (in addition to standard therapy) significantly reduced morbidity and mortality in heart failure with reduced ejection fraction (HFrEF) patients [65]. However, the later TOPCAT clinical trial [67] showed that patients with heart failure and preserved ejection fraction (HFpEF) showed no significant improvements with the addition of spironolactone to traditional therapy. We have previously shown that the Cp1a1Ren2 hypertensive rat model (the control groups in this study) best fits with HFpEF [68] while it is recognised that the LAD surgical model best fits with HFrEF. Therefore, this rodent model offers a unique insight into both of these pathophysiological disorders and the current study demonstrated significant improvements in both HFpEF and HFrEF as well as renal fibrosis.

## Conclusion

Using a transgenic inducible hypertensive Ren2 rat model to first establish hypertension before superimposing the common LAD surgical model of myocardial infarction, we created a

progressive model of cardiorenal dysfunction more closely representing that seen in a clinical setting. The addition of a clinically relevant dose of spironolactone post MI, blunted the progression of cardiac and kidney inflammation and fibrosis. This was despite no reduction in systolic blood pressure. While the exact mechanisms of spironolactone for these actions are beyond the scope of this study, they are clearly multifaceted, and, in part, blood pressure independent. Further work is planned to categorise the potential mechanisms that may mediate the actions of spironolactone.

## Methods

### Animals

The initial transgenic Cyp1a1Ren2 rat internal breeding stock was gifted by Professor J.J. Mullins (Centre for Cardiovascular Science, University of Edinburgh, UK). The colony was held at the University of Otago Animal Resource Unit. Animals were housed under controlled conditions of temperature (~21°C) and light (12-h light/dark cycle), with food (meat-free rat and mouse diet, irradiated, Specialty Feeds, Australia) and tap water provided *ad libitum*. Male Cyp1a1-Ren2 rats used for experiments were obtained from internal breeding stock and housed in groups. All experiments were approved by the Animal Ethics Committee of the University of Otago (AEC 51/13) and in accordance with the guidelines of the New Zealand Animal Welfare Act [69].

The Cyp1a1Ren2 rat allows hypertension to be reversibly induced by addition of an aryl hydrocarbon to the diet, without the need for surgical intervention [70]. An inducible cytochrome p450-1a1 promoter controls the expression of mouse Ren2 cDNA integrated into the Y chromosome of Fischer 344 rats [70,71]. Administration of dietary indole-3-carbinol (I3C) leads to activation of the promoter gene (Cyp1a1), resulting in activation of the inserted Ren2 gene [70,72,73], driving increased circulating renin levels (primarily in the liver), activating the renin-angiotensin-aldosterone system (RAAS) and resulting in a consequent increase in blood pressure. Significantly, the extent of hypertension is also I3C dose dependent [71,74,75], allowing for tight titration of blood pressure.

The experimental overview is presented in Fig 3. Experimental animals (n = 26) were randomised to groups at the beginning of the experiment: a hypertensive group (H; n = 4), a

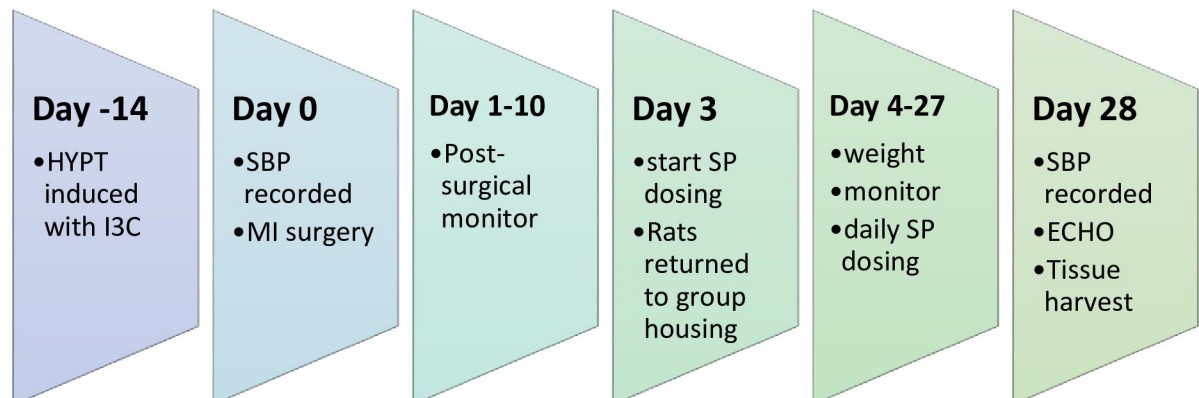

**Fig 3. Experimental overview.** Hypertension (HYPT) was induced (and maintained) with a chow containing 0.167% w/w indole-3-carbinol (I3C) for two weeks prior to the start of the experiment. On day one of the experimental procedure, animals had their systolic blood pressure (SBP) recorded and underwent the myocardial infarction (MI) surgery. The following 10 days the animals post surgically monitored closely (including weight, coat and wound condition and overall health). On experimental day 3, animals were repaired into group housing and began daily oral dosing (by syringe feeding) of spironolactone (SP). On experimental day 28, animals SBP was recorded and an echocardiogram (ECHO) performed, before the animals were terminated and the tissues harvested for analysis.

hypertensive group treated with daily spironolactone (H+SP; n = 4), a hypertensive group with surgically induced myocardial infarction (H+MI; n = 6), and subsequently a hypertensive group with surgically induced myocardial infarction and treated with daily spironolactone (H+MI+SP; n = 6). An additional hypertensive group (n = 6) underwent surgery—thoracotomy but where no cardiac ligation was placed (Sham group).

**Chronic elevation of blood pressure and systolic blood pressure recording.** Hypertension was established over two weeks (from 8 to 10 weeks of age) in Cyp1a1Ren2 rats, as previously reported [75], by feeding irradiated pelleted standard chow with added 0.167% indole-3-carbinol w/w (I3C, #SF13-086, Specialty Feeds, Perth, Australia) to activate and maintain hypertension.

Systolic blood pressure (SBP) was measured using tail-cuff plethysmography (NIBP controller plus PowerLab 4SP, ADInstruments, Dunedin, New Zealand) on Days 0 and 28. Data was captured and analysed using Chart v.7 software (ADInstruments, Dunedin, New Zealand). A minimum of eight clear recordings were taken from each rat on each occasion.

**Surgical induction of a myocardial infarction.** All surgery was performed using aseptic techniques and performed in accordance with ethics and welfare regulations. Myocardial infarction was surgically induced by ligating the left anterior descending artery (LAD). Animals were given Dormotor® (0.2mg/kg, s.c), ventilated mechanically and anaesthetic administered (Halothane; 2% at induction, reducing to 0.9–1% for maintenance, with 70:30 $O_2/N_2O$). A left thoracotomy was performed between the second and third intercostal space and a permanent ligature (7–0 prolene) was placed around the left coronary artery. After the onset of ischaemia, the thoracic incision was closed. The perioperative mortality in the first 24h was 17%. The sham surgical group underwent the same surgery but where no LAD ligature was placed.

**Spironolactone dosing.** Dosing was adjusted using the allometric scaling calculations as described by Reagan-Shaw *et al* [59]. A human equivalent dose of 50mg per day of spironolactone (Sigma-Aldrich, Missouri, USA) was used, equating to a dose of 4.41mg/kg/day for a rat. To enhance acceptance, spironolactone was mixed into a caramel syrup (Quaterpast, Shott Beverages Ltd., Auckland, New Zealand). Dosing began on day three of the experimental period.

**Urine analysis.** On Day 26, animals were weighed and placed into single housed metabolic cages (Techniplast, PA, USA) for 12 hours (7am to 7pm). Total urine volume was recorded, the urine spun (1500rpm for 15 minutes), and aliquots stored at -80˚C for later analysis.

Urinary protein concentration was measured in triplicate using a BCA assay (Pierce BCA protein assay kit, Thermo Scientific, USA). Urine creatinine was measured using a Cobas c310 (Roche, USA) with a CREJ2 creatinine Jaffe Gen.2 kit (Cobas, Roche, USA) and results are expressed as a protein to creatinine ratio.

**Serum creatinine.** Blood was collected via cardiac puncture at termination and centrifuged at 1500rpm for 15 minutes at 4˚C to collect serum. Serum creatinine was determined in triplicate for each animal (Randox CR510, UK).

**Echocardiography.** Echocardiography was performed on Day 28 as previously described [68]. Briefly, rats were placed supine and transthoracic echocardiography was conducted using a 10MHz linear probe (GE ML6-15, GE Healthcare, Chicago, USA) connected to a standard echocardiography system (Vivid 9, GE Healthcare, Chicago, USA). At least three consecutive cardiac cycles of standard two-dimensional long- and short-axis (at the mid-papillary level) images were acquired before being transferred for offline analysis using an image analysis package (2D CPA, TomTec Image-Arena, version 2.21; TomTec Imaging Systems, Unterschleissheim, Germany). Determination of LV ejection fraction (LVEF) and LV end volumes were performed using Simpson's method on TomTec Image-Arena.

## Quantitative microscopy

**Heart.** Fixation and quantification of the heart followed methods previously described [68]. In brief, hearts were removed following anaesthetic overdose and arrested using 15mmol. $L^{-1}$ potassium chloride in Hartman's saline. Hearts were fixed for four hours in 10% neutral buffered formalin (NBF), before transversely cut into 3mm sections (measuring from the apex) and further fixed in 10% NBF overnight at room temperature. Tissue sections were dehydrated by passage through alcohols and embedded in paraffin wax and cut at 5μm. Tissue was stained, using standard procedure, with picrosirius red with light green counter stain and mounted in DPX.

Stained sections were viewed using a Zeiss Axioplan Microscope (Zeiss, Oberkochen, Germany), and images of representative regions were recorded using a Nikon microscope camera (DS-Ri2, Nikon, Tokyo, Japan). Cardiac interstitial fibrosis was quantitatively assessed using Sirius red, capturing a minimum of 10 non-overlapping, evenly distributed, myocardial sections (x100 magnification), containing no vessels, from the inferior cardiac segment (well away from the area of infarction), taken 6mm from the apex from each animal. The extent of fibrotic tissue was quantified by applying a trained pixel classifier to each section (NIS Elements Basic Research Imaging software, Version 5.11 (64bit edition), Nikon, Tokyo, Japan), giving a percentage of the positively stained pixels (fibrotic tissue). For each individual animal, a series of 10–20 fields at high magnification were used (depending on the stain analysed), with the percentage of positive pixels for each field recorded (giving a minimum of 40 data points for each group). Statistical comparisons between groups were accomplished by one-way analysis of variance (ANOVA) with Bonferroni post-hoc analysis using GraphPad Prism.

**Size of Myocardial infarction.** The size of each myocardial infarction was calculated using a geometric angle calculation, as outlined by Lichtenauer and colleagues [76]. In short, the centroid of the left ventricle was calculated using image analysis software (NIS Elements imaging software–Basic research, Nikon, Japan) for each of the transverse sections (3mm, 6mm and 9mm from the apex). Angle calculations were performed in relation to endo-, myo- and epi-cardial border zones between vital myocardium and scar tissue, using the image software (NIS Elements imaging software). Infarct size (or angle) was calculated as the mean of all three cardiac layer measurements and expressed as a percentage of the total area of the left ventricle.

**Kidney.** Fixation and quantification of the kidney followed methods previously described [56]. In brief, a transverse 4-5mm section was taken from the centre of the right kidney, fixed in 10% NBF and microwaved (55˚C, 250 watts) for 5 minutes. Sections were left in 10% NBF at room temperature overnight before being dehydrated by passage through alcohols and embedding in paraffin wax. Kidney tissue was cut at 3μm and stained with either picrosirius red (SR) or periodic acid-Schiff (PAS), followed by mounting in DPX. Stained sections were viewed using a Zeiss Axioplan Microscope (Zeiss, Germany), and images of representative regions were recorded using a Nikon microscope camera (DS-Ri2, Nikon, USA) and Nikon proprietary software (NIS Elements Basic Research imaging software, Nikon, USA).

Renal cortical fibrosis was quantitatively assessed using SR by capturing a minimum of 10 serial, non-overlapping regions (x50 magnification), containing no blood vessels, across the mid renal cortex from each animal. The extent of fibrotic tissue was quantified by applying trained pixel classifier software (NIS Elements Basic Research Imaging software, Nikon, Japan) to each region and expressed as a percentage.

**Glomerulosclerosis Index (GSI).** Glomerulosclerosis was assessed following methods previously described [56]. In short, a renal cortical section stained with PAS, was scanned transversely at x100 magnification, began at a randomly chosen site, examining 50 glomeruli. The degree of glomerulosclerosis was assessed using a semi quantitative scoring method

previously published [56], to give a glomerulosclerosis index (GSI). Scoring was performed in a double-blinded manner by three experienced independent observers.

**Immunohistochemistry.** Antibodies (diluted in 1% Bovine serum albumin (BSA, Sigma, fraction V, USA)) used were against alpha smooth muscle actin (αSMA, 1:50, monoclonal, A5228, Sigma-Aldrich, USA), and Cluster of Differentiation 68 (CD68, 1:100, MA5-13324, monoclonal, Thermofisher, USA). Negative controls were carried out either by omitting the primary antibodies and/or by using appropriate blocking peptides. All analysis was performed in a blinded manner and cross checked by a second, experienced, blind observer.

Antigen retrieval was carried out (microwave for 10 minutes in 10mmol/L citrate buffer at pH 6.0), with the exception of the αSMA antibody, followed by blocking of endogenous peroxidase activity with 3% $H_2O_2$ in PBS. Sections were then preincubated in 1% BSA (Sigma-Aldrich, MO, USA) in PBS to block nonspecific binding, before labelling with the appropriate antibody overnight. Antibodies were visualised using an appropriate horseradish peroxidase-coupled secondary antibody (anti-mouse IgG or anti-goat IgG, 1:25, Dako, Denmark) followed by incubation with 3,3-diaminobenzidine substrate (SigmaFast tablets, Sigma-Aldrich, USA) and counter stained with Ehrlich's haematoxylin. After dehydration and clearing, sections were mounted in DPX.

## Quantification of immunohistochemistry

**αSMA.** For cardiac assessment, 20, non-overlapping fields at x100 magnification were taken. Sampling began at a randomly chosen site on the boarder of the inferior segment and the section was scanned transversely, ensuring each chosen field contained as few vessels as possible. Captured images were assessed (using the NIS Elements Basic Research imaging software) by counting the number of positive stained aSMA cells and was expressed as myofibroblasts/mm$^2$.

In the renal cortex, a semi quantitative score was adapted from that used by Johnson *et al* [77], and selections assigned to the tubular interstitial fibroblast proliferation in each captured renal selection, where: 0—No interstitial fibroblasts, 1 - >10 fibroblasts per field, 2–10–100 fibroblasts per field, 3–1–5 tubules encased in fibroblasts, 4—Many tubules and glomeruli totally encased by fibroblasts. Ten fields (x50 magnification), containing as few vessels as possible, were examined from each animal and a mean score was calculated.

**CD68.** A minimum of 10 fields were captured (x50 magnification) for each animal (cardiac and renal). Positively stained cells were counted (using the NIS Elements Basic Research imaging software) and expressed as macrophage/mm$^2$.

**Statistics.** Data is presented as means ± standard deviation. Statistical comparisons were accomplished by one-way analysis of variance (ANOVA) with Bonferroni post-hoc analysis using GraphPad Prism (GraphPad software, Inc. version 5.03). Correlations were performed using Pearson's correlation (GraphPad Prism, GraphPad software, Inc. version 5.03). Results were considered to be statistically significant if P values were <0.05.

## Supporting information

**S1 Data.**
(XLSX)

## Acknowledgments

We would like to acknowledge the use of facilities provided by A.Prof I. Sammut. We would also like to acknowledge the assistance in the preparation of tissue samples by M. Fisher at the Otago University Histology Unit.

## Author Contributions

**Conceptualization:** R. J. Walker.

**Data curation:** C. J. Leader.

**Formal analysis:** C. J. Leader, G. T. Wilkins, R. J. Walker.

**Funding acquisition:** R. J. Walker.

**Investigation:** C. J. Leader, G. T. Wilkins, R. J. Walker.

**Methodology:** C. J. Leader, G. T. Wilkins.

**Project administration:** C. J. Leader, G. T. Wilkins, R. J. Walker.

**Resources:** C. J. Leader.

**Supervision:** G. T. Wilkins, R. J. Walker.

**Validation:** C. J. Leader.

**Writing – original draft:** C. J. Leader, R. J. Walker.

**Writing – review & editing:** C. J. Leader, G. T. Wilkins, R. J. Walker.

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
