## [Decision Letter · Decision Letter 0]

19 Oct 2021

PONE-D-21-28323The effect of spironolactone on cardiac and renal fibrosis following myocardial infarction in established hypertension in the transgenic Cyp1a1Ren2 rat.PLOS ONE

Dear Dr. Walker,

Thank you for submitting your manuscript to PLOS ONE. After careful consideration, we feel that it has merit but does not fully meet PLOS ONE’s publication criteria as it currently stands. Therefore, we invite you to submit a revised version of the manuscript that addresses the points raised during the review process.

 All issues raised by expert reviewers are required.

We look forward to receiving your revised manuscript.

Kind regards,

Vincenzo Lionetti, M.D., PhD

Academic Editor

PLOS ONE

Journal Requirements:

2. Please ensure that all data are included in the manuscript rather than stating 'data not shown' (i.e. line 94)

We note that you have included the phrase “data not shown” in your manuscript. Unfortunately, this does not meet our data sharing requirements. PLOS does not permit references to inaccessible data. We require that authors provide all relevant data within the paper, Supporting Information files, or in an acceptable, public repository. Please add a citation to support this phrase or upload the data that corresponds with these findings to a stable repository (such as Figshare or Dryad) and provide and URLs, DOIs, or accession numbers that may be used to access these data. Or, if the data are not a core part of the research being presented in your study, we ask that you remove the phrase that refers to these data.

3. Please ensure that you refer to Figure 1 in your text as, if accepted, production will need this reference to link the reader to the figure.

Reviewers' comments:

Reviewer's Responses to Questions

**Comments to the Author**

1. Is the manuscript technically sound, and do the data support the conclusions?

Reviewer #1: Partly

Reviewer #2: Yes

2. Has the statistical analysis been performed appropriately and rigorously? 

Reviewer #1: I Don't Know

Reviewer #2: Yes

3. Have the authors made all data underlying the findings in their manuscript fully available?

Reviewer #1: Yes

Reviewer #2: Yes

4. Is the manuscript presented in an intelligible fashion and written in standard English?

Reviewer #1: Yes

Reviewer #2: Yes

5. Review Comments to the Author

Reviewer #1: This is an interesting study and the following points need clarity:

1. There is a very noticeable reduction in infract size (Fig 2a) with drug treatment, yet functional recovery of myocardium is modest at best. Can the authors address this please.

2. More clarity on the quantification of data from immuno-staining sections and stats/programs used here would help.

3.What is specificity of anti-alpha SMA? Would these be smooth muscle cells....ie could this vascular remodeling?

4. Is there hypertrophy of glomeruli in Fig 3 (top panel) ?

5. Hypertrophy of myocardium after MI and impact of drug should be addressed.

Reviewer #2: This study "examined the actions of mineralocorticoid receptor antagonists on progressive cardiac injury and kidney injury after myocardial infarction in an animal model of established hypertension". The study has important implications for future clinical research, as many patients with myocardial infarction are hypertensive and current ESC guidelines recommend a mineralocorticoid receptor antagonist only for patients developing systolic dysfunction during the acute phase of the infarction. On the other hand, I have several concerns with study design and presentation of results. Please find some specific comments below.

A "mortality rate of just 17% despite the average infarct size being approximately 40%" does not seem plausible.

Human patients with hypertension and/or myocardial infarction usually receive an ACE inhibitor or an angiotensin receptor blocker, while your rats received at best spironolactone only. Under this respect, your model did not replicate the clinical phenotype.

Patients with myocardial infarction usually undergo coronary reperfusion, while your model involved a permanent coronary ligation. Why did you choose permanent over temporary ligation?

The site of coronary ligation was clearly not standardized, therefore infarct size was likely highly heterogeneous between animals. You evaluated absolute infarct size without considering this element.

Cardio-renal syndrome is repeatedly mentioned, although the study did not directly evaluate the heart-kidney interaction, but rather the effects of spironolactone on both heart and kidney.

Please justify the number of animals in each group. Given the small number, distribution was most likely non-normal; the use of mean +/- standard deviation is therefore not optimal.

In the Abstract, please add the number of animals in each group. You should also specify whether animals underwent reperfusion after coronary occlusion or not. Furthermore, "metabolic studies" is rather vague; even after reading the whole paper I do not understand which are these studies.

6. PLOS authors have the option to publish the peer review history of their article (what does this mean?). If published, this will include your full peer review and any attached files.

Reviewer #1: No

Reviewer #2: No

---

## [Author Response · Author response to Decision Letter 0]

3 Nov 2021

We would like to thank the editor and both reviewers for their helpful comments and suggestions in regard to the current manuscript titled “The effect of spironolactone on cardiac and renal fibrosis following myocardial infarction in established hypertension in the transgenic Cyp1a1Ren2 rat.”

Comments from the Editor

Please ensure that all data are included in the manuscript rather than stating 'data not shown' (i.e. line 94)

We note that you have included the phrase “data not shown” in your manuscript. Unfortunately, this does not meet our data sharing requirements. PLOS does not permit references to inaccessible data. We require that authors provide all relevant data within the paper, Supporting Information files, or in an acceptable, public repository. Please add a citation to support this phrase or upload the data that corresponds with these findings to a stable repository (such as Figshare or Dryad) and provide and URLs, DOIs, or accession numbers that may be used to access these data. Or, if the data are not a core part of the research being presented in your study, we ask that you remove the phrase that refers to these data.

Full data has been added into the text for reference 

“Sham surgery resulted in no significant measured physiological differences from the hypertensive control animals (SBP (179±8mmHg vs 174±11mmHg), EF% (77±2% vs 82±2%), EDV (0.31±0.04ml vs 0.37±0.02ml), ESV (0.07±0.01ml vs 0.07±0.01), Na/K urinary ratio (0.63±0.2 vs 0.55±0.03), protein/creatinine ratio (17.2±2.8 vs 15.5±3.9) and plasma creatinine (107.2±9.1µmolL-1 vs 95.8±11.4 µmolL-1)).” (page 5 lines 93 -97)

3. Please ensure that you refer to Figure 1 in your text as, if accepted, production will need this reference to link the reader to the figure.

This figure is an overview of the experimental studies. It is now referred to in the methods section (page 14 line 307) and has been renumbered as figure 3. Figures 2 & 3 have been renumbered as figures 1 & 2. These have been changed in the text as well. 

Reviewer 1:

1. There is very noticeable reduction in infarct size (fig 2a) with drug treatment, yet functional recovery of myocardium is modest at best. Can the authors address this please. 

Thank you for raising this. There are several possible explanations for this. As a consequence of the surgery (the opened pericardium was not able to be closed), the anterior wall of the heart ended up being adherent to the interior rib cage by four weeks (when animals were terminated). This adherence almost certainly affected and limited some of the functional properties of the heart in all the animals given the myocardial infarction. Equally due to the large size of the infarctions there is still substantial loss of the anterior wall of the left ventricle which will also clearly impact upon function. The adherence to the anterior chest wall also limited the extent of echo assessment of left ventricular function. Also further image angles are not possible to obtain in a rodent model, ruling out the use of Simpsons rule. 

In this model, Spironolactone did not substantially lower systolic blood pressure, and the prolonged hypertension would have still promoted ongoing left ventricular hypertrophy in the unaffected myocardium. There is not unexpected that the cardiac functional parameters did not show an improvement, despite a reduction in infarction size. 

We have added a section to the discussion about this limitation. Page 12 lines 274 -278.

2. More clarity on the quantification of data from immune-staining sections and stats/programs used here would help

For the immunohistochemical staining, assessments were performed using pixel classification software and an inbuilt trainable algorithm (NIS Elements Basic Research imaging software, Nikon). Using the Nikon software, the algorithm is first ‘trained’, before the percentage of total pixels positively stained can be calculated. Training the software, consists of assigning individual pixels (at very high magnification) from an image into either positive or negative staining, starting with pixels that are strongly positive or negative. These manual selections aid the software to ‘learn’ and assign pixels to a selected group (in this case positive or negative stain). For the current study, this training was performed using 5-8 images for each individual animal, and a minimum of 1000 manually assigned pixels. Before running the trained algorithm, a trial of the test images are used to ensure a good match to the IHC stain. Once completed, all the images of each individual animal were run through the software and the output recorded as a percentage of positive pixels for each selected image/field. 

For each individual animal, a series of 10-20 fields at high magnification were used (depending on the stain analysed), with the percentage of positive pixels for each field recorded (giving a minimum of 40 data points for each group). Statistical comparisons between groups were accomplished by one-way analysis of variance (ANOVA) with Bonferroni post-hoc analysis using GraphPad Prism.

This last paragraph has been added to the methods section for further clarification. Page 17 lines 394 -399.

3. What is specificity of anti-aSMA? Would these be smooth muscle cells…ie could this vascular remodelling?

SMA is often used as a marker of myofibroblasts. Myofibroblasts are activated cells that express the most extracellular matrix genes in the setting of injury. They are usually thought of as activated fibroblasts but there is increasing evidence that they are derived from activated stromal cells. In the kidney, transformed pericytes are thought to be the most likely source of myofibroblasts associated with interstitial fibrosis. The exact origin of the cardiac myofibroblasts is not clear but are certainly associated with activated fibroblasts. They are not derived from smooth muscle cells. There is no good evidence to support epithelial to mesenchymal transformation (1). While we cannot rule out vascular remodelling contributing to the presence of increased �SMA positive cells, given that the semi-quantitative scoring was in regions well away from any vessels we feel that this is unlikely.

The semi-quantitative method used for scoring aSMA has been used widely by the Walker laboratory (for example Walker et al [2] and Kalita-De Croft et al [3]), and was based on the methodologies used by Johnson et al [4] and others [5-7]. While we also accept that IHC can be relatively inaccurate when the distribution is discontinuous, we ensured that multiple measurements were made from each tissue section as described in the text (Methods). While it would have been desirable to confirm the results by Western blotting and other methods, in these cases the antibodies available were not suitable for both IHC and Westerns.

4. Is there hypertrophy of glomeruli in Fig 3 (top panel)?

In short, yes. All of the animals in the present study were hypertensive throughout the study. This has also been previously documented for this strain [8–11]. Glomerular hypertrophy occurs with hypertension and focal glomerulosclerosis. Injury to some glomeruli induce a compensatory hypertrophied response in remaining glomeruli leading to further glomerular injury and progressive glomerular sclerosis. This is a well-established paper of progressive kidney injury. In figure 2, glomerulosclerosis in all groups is significantly increased from normotensive animals (noted by the red line), suggesting that hypertension alone can cause significant renal injury, although this possibility was not perused further. While hypertrophy of the glomeruli was observed, the degree of glomerulosclerosis is a more critical marker of kidney injury. This is why we have reported the glomerulosclerosis index and it’s correlation with interstitial fibrosis (both cardiac and kidney fibrosis) 

5. Hypertrophy of myocardium after MI and impact of drug should be addressed.

Please see the response to point 1 above as this point overlaps with point 1. We agree that assessment of hypertrophy of the myocardium would be nice to assess. However, this was not done due to a number of difficulties. 

In order to assess over all cardiac hypertrophy, we would need further critical ECHO assessments. Further image angles are not possible to obtain in a rodent model, ruling out the use of Simpsons rule. Equally due to the large size of the infarctions, limited strain assessments were also not possible. Of note, due to the high level of connective tissue causing adherence between the heart and the rib cage, assessments of hypertrophy of the heart were also compromised. 

With our histological assessments, we did investigate the use of LV wall measurements. However, again due to the large size of the infarctions, there were limited places to conduct these measurements (with only the mid inferior-septal wall consistently free from infarction). Despite this we did attempt this, but found no significant differences in wall thickness between these groups. 

Lastly, all of the animals within the present study were hypertensive. Spironolactone resulted in no significant decrease to systolic blood pressure. Our assumption is that one of the key driving forces for cardiac hypertrophy is the underlying and continued high SBP. It would however be an interesting point to investigate if the addition of spironolactone did have some protective effect against such mechanical stressors. 

Reviewer 2:

1. A “mortality rate of just 17% despite the average infarct size being approximately 40%” does not seem plausible. 

Six animals were designated into each of the groups that underwent the difficult and delicate MI surgery. It is also worth noting that these animals were of a smaller size than is commonly reported for this surgery which may have played a positive role in survival. The restrictions of numbers of animals within each group was governed by the University ethics committee, in part due to our research groups previously proven high survival rate (>75%) allowing for fewer numbers to be used and equally to adhere to the 3Rs of animal research ethics. The first author is a very experienced and capable small animal handler. To aid in increasing survival outcomes, a number of measures were taken. The animals were put through several weeks of handling to reduce stress and ensure they were comfortable with the researcher and handling procedures. Animals were also always given a minimum of 30 minutes of acclimatization following any movement between areas. All of the surgery was performed by the first author, and she essentially ran the equivalent of a ‘rat post cardiac surgery ICU’ so each animal was closely and constantly individually observed for minimum of 6 hours post-surgery to prevent any peri-operative arrest which could have happened. The results reported are definitely true and reflect the extensive peri-operative and post-operative care provided to these animals. Furthermore, all animal mortality had to accounted for and was reported to the university’s veterinary service who then review every case. So these results can be validated by our veterinary service if required.

Further, for surgery, we used a slight variation in the anaesthetics and analgesics (Midazolam and Dormator in addition to halothane gas) and extended oxygen tent therapy using carbogen (minimum of four hours). We believe that these measures enabled a slower recovery from anaesthetics out of surgery (and limited movement), greatly reducing stress both pre- and post-surgery and improved oxygen handling while in recovery. In turn, taking these precautions, alongside highly trained and careful surgery, enabled our group to have a high survival rate with only two from 12 animals dying within six hours of surgery. While we acknowledge that the recognised and accepted mortality rate for this surgery is 50%, survival rates of greater than 75% have also been published [12,13]. However, there are also many publications utilizing this surgery do not publish the numbers of animals used nor the survival/mortality rates [13–15]. 

The infarct sizes published are large but are also in line with what many published studies in rats have shown. In fact, the anatomical position chosen to ligate was actually lower on the coronary artery than many other authors suggest or perform. It is commonly stated for ligament placement to be as close as possible to the branching from the circumflex artery [16–19]. Earlier studies by our research group, found that in this rat strain, the anterior coronary artery runs slightly more posterior than traditionally reported in other rat strains (such as the Sprague Dawley rat). 

2. Human patients with hypertension and/or myocardial infarction usually receive an ACE inhibitor or an angiotensin receptor blocker, while your rats received at best spironolactone only. Under this respect, your model did not replicate the clinical phenotype.

Large clinical studies such as RALES [20] have demonstrated the efficacy of mineralocorticoid receptor antagonism in patients with heart failure showing that low dose mineralocorticoid receptor antagonism (in addition to standard therapy) significantly reduced morbidity and mortality in heart failure with reduced ejection fraction (HFrEF) patients [20]. Additionally, while several animal studies have shown that spironolactone reduces cardiac fibrosis and improves cardiac dynamics across a variety of models and settings, they have utilized rather high translational doses, followed a short duration, or used poor methodology. The aim of this study was therefore to investigate the pathophysiological modifications with monotherapy of clinically relevant doses of spironolactone. Additionally, we aimed to mimic the clinical setting or order of damage – hypertension, followed by cardiac injury and associated renal injury prior to treatment. 

The focus of our study was the effects of mineralocorticoid receptor antagonist function on cardiac remodelling post infarct and also how this modified kidney injury. We accept that ACEI or ARB are also utilised in this clinical setting. However, it is important to document the actions of each individual agent. Provided we can obtain additional funding we certainly plan to repeat these studies with an ACEI or ARB separately then combined to further tease out the effects. At this stage we are reporting the effects of MRA alone

3. Patients with MI usually undergo coronary reperfusion, while your model involved permanent coronary ligation. Why did you choose permanent over temporary ligation? 

This surgery is high risk. Temporary ligation would have been much more difficult to maintain the animal alive peri-operatively and raises questions as to what time course to reverse the ligation would be most relevant. We wanted to established myocardial infarction in this model, as unfortunately there are still large numbers of people who present late after their myocardial infarction where reperfusion is not feasible. The subsequent effects of the myocardial infarction injury both on cardiac function and renal function was the focus of this study.

The most commonly used animal model for MI in vivo is surgical ligation of the left descending coronary artery (LAD), with two predominant approaches: permanent ligation or ischaemia-reperfusion (where the LAD is temporarily occluded before removing the suture to restore bloody flow). Permanent ligation is often chosen to be a more appropriate model for studies of heart tissue injury and wound healing, owing to larger and more consistent infarcts than the ischaemia-reperfusion model. 

We aimed initially to develop a model of cardio renal syndrome (both in a normotensive and hypertensive animal model) utilising cardiac damage (LAD infarction) as the major driving force. To replicate this, we required significant, permanent and chronic injury to the heart tissue and therefore chose the permanent ligation of the LAD for this. This permanent cardiac injury allowed us to develop a model in which subsequent renal injury was apparent (even in normotensive animals) and establishing the cardiorenal connection for our studies. 

4. The size of coronary ligation was clearly not standardised; therefore, infarct size was likely highly heterogeneous between animals. You evaluated absolute infarct size without considering this element. 

Myocardial infarctions do not occur in uniform size and fashion in a clinical setting. It is equally almost impossible to obtain a uniform infarction across multiple individuals due to a number of factors and therefore it is also expected to be heterogeneous between animals. We did attempt to standardise the procedure wherever possible. While one can aim to limit some of the factors associated with this complex surgery completed in a small operating field, things such as exact suture depth, use of biological landmarks and placing the ligature by hand into a small, fast-beating heart, will inevitably create some differences between individuals. This is an accepted limitation of the myocardial infarction LAD surgery. 

To attempt to standardise an infarct size across all animals in this study, infarct sizes had to be involving over 20% of the LV and must have extended beyond 6mm from the apex to be included in the study. All surviving animals fitted these limited parameters. 

Each animal had the infarct size calculated individually, but in order to form some consistency, a transverse section of set distance from the apex (6mm) was selected to assess. Previous published work [21] from our group used several methods to ensure that the infarct size inferred from the 6mm transverse section (using previously established published methods) gave a similar result to assessing the entire affected infarction area as a percentage of the LV[21,22]. 

By measuring the absolute infarct size, we are providing the reviewers with an accurate measure of the infarct along with echo data of function. In all studies there will be some variability of the infarct so the mean and variation have been provided This variation is unavoidable, but equally does also represent the fact that in the clinical setting there will also inevitably be variation in infarction size between individuals. 

5. Cardiorenal syndrome is repeatedly mentioned, although the study did not directly evaluate the heart -kidney interaction, but rather the effects of spironolactone on both heart and kidney. 

Cardiorenal syndrome is defined as “acute or chronic dysfunction in one organ may induce acute or chronic dysfunction of the other”[23]. 

We have reported data on kidney function and cardiac function along with measurements of blood pressure in control animals, hypertensive animals without an infarct and hypertensive animals with an infarct. There are clear differences in both cardiac and kidney function with a close correlation between hypertension, function and fibrosis as well as the added impact of MI on kidney function and fibrosis. See figure 2 that demonstrates the correlations between cardiac and renal fibrosis as well as glomerulosclerosis. We also report how these parameters were modified by the actions of spironolactone. So we believe we have described the interactions between kidney and heart in this setting. 

6. Please justify the number of animals in each group. Given the small number, distribution was most likely non-normal; the use of mean ± Std dev is therefore not optimal. 

This research utilized n=4 to 6 for each experimental group. We acknowledge and understand that the use of the low number of animals per group is a valid concern. However there were several considerations and constraints that restricted the animal numbers allowed by our Ethics Committee for this study. The Otago University Animal Ethics committee required very strong justification for not only the use of the animals, but also as to the total numbers requested with the requirement from the Ethics committee to use as few animals as possible. 

The rat strain (Cyp1a1Ren2 rat) is a highly inbred, transgenic strain, making the individual animals genetically homogeneous, and permits reduced numbers to be used (essentially no individual variance so we also expect distribution to be normal). Initially we aimed to have a total number of 8 animals per group. With no previously published data available, a power calculation to define the number required was not possible. Therefore, to calculate the sample size required the “resource equation” method was used [24]. Using this equation, numbers of four or greater was deemed to be significant. Our decision was justified post hoc, as the variance determined between individuals within each group was within two standard deviations from the mean. Further, significant results were obtained with these numbers. Also similar studies, using this model, have published significant data using only 3 to 5 animals in the different experimental groups [25,26]. 

In the tabulated data, the mean with standard deviation is provided, alongside the 95% confidence intervals for that data. Statistical analysis was also performed to establish if there was any significance difference across groups (with probability set at the standard 0.05). All statistics performed in both this and previous similar published studies by this group, were done alongside a qualified statistician from the University of Otago. 

7. In the abstract, please add the number of animals in each group. You should also specify whether animals underwent reperfusion after coronary occlusion or not. Furthermore “metabolic studies” is rather vague; even after reading the whole paper I do not understand which these are. 

We have added numbers to the abstract and clarified that it was a permanent ligature to induce the MI. In animal studies, placement in metabolic cages for overnight urine collections followed by blood samples is usually referred to as metabolic studies. To avoid confusion, this has been redefined in the abstract as biochemical testing. The data obtained is presented in table 1. 

1. Fogo A. Searching for the origin of the myofibroblasts one cell at a time. Kidney Int. 2021; 99: 1259 - 61

2. Walker RJ, Leader JP, Bedford JJ, Gobe G, Davis G, Vos FE, et al. Chronic interstitial fibrosis in the rat kidney induced by long-term (6-mo) exposure to lithium. Am J Ren Physiol. 2013;304: F300-307. 

3. Kalita-De Croft P, Bedford JJ, Leader JP, Walker RJ. Amiloride modifies the progression of lithium-induced renal intersitial fibrosis. Nephrology. 2018;23: 20–30. 

4. Johnson RJ, Alpers CE, Yoshimura A, Lombardi D, Pritzl P, Floege J, et al. Renal injury from angiotensin II-mediated hypertension. Hypertension. 1992;19: 464–474. 

5. WANG M, LIU R, JIA X, MU S, XIE R. N-acetyl-seryl-aspartyl-lysyl-proline attenuates renal inflammation and tubulointerstitial fibrosis in rats. Int J Mol Med. 2010;26: 795–801. 

6. Li B, Tang Y, Wang W, Xie Y, Wang N, Yuan Q, et al. Fluorofenidone attenuates renal interstitial fibrosis in the rat model of obstructive nephropathy. Mol Cell Biochem. 2011;354: 263–273. 

7. Yang N, Wu LL, Nikolic-Paterson DJ, Ng YY, Yang WC, Mu W, et al. Local macrophage and myofibroblast proliferation in progressive renal injury in the rat remnant kidney. Nephrol Dial Transplant. 1998;13: 1967–1972. 

8. Kantachuvesiri S, Fleming S, Peters J, Peters B, Brooker G, Lammie AG, et al. Controlled hypertension, a transgenic toggle switch reveals differential mechanisms underlying vascular disease. Jounral Biol Chem. 2001;276: 36727–36733. doi:10.1074/jbc.M103296200

9. Peters B, Grisk O, Becher B, Wanka H, Kuttler B, Ludemann J, et al. Dose-dependent titration of prorenin and blood pressure in Cyp1a1ren-2 transgenic rats: absence of prorenin-induced glomerulosclerosis. J Hypertens. 2008;26: 102–109. doi:10.1097/HJH.0b013e3282f0ab66\\r00004872-200801000-00017 [pii]

10. Mullins JJ, Peters J, Ganten D. Fulminant hypertension in transgenic rats harbouring the mouse Ren-2 gene. Nature. 1990. pp. 541–4. doi:10.1038/344541a0

11. Heijnen BFJ, Nelissen J, van Essen H, Fazzi GE, Cohen Tervaert JW, Peutz-Kootstra CJ, et al. Irreversible Renal Damage after Transient Renin-Angiotensin System Stimulation: Involvement of an AT1-Receptor Mediated Immune Response. PLoS One. 2013;8: e57815. doi:10.1371/journal.pone.0057815

12. Wang,Hao, Huang,Bing S, Ganten D. Prevention of Sympathetic and Cardiac Dysfunction After Myocardial Infarction in Transgenic Rats Deficient in Brain Angiotensinogen. Circ Res. 2004;94: 843–849. doi:10.1161/01.RES.0000120864.21172.5A

13. Segersvard H, Lakkisto P, Forsten H, Immonen K, Kosonen R, Palojoki E, et al. Effects of angiotensin II blockade on cardiomyocyte regeneration after myocardial infarction in rats. J Renin-Angiotensin-Aldosterone Syst. 2013;16: 92–102. doi:10.1177/1470320313487567

14. Wiemer G, Itter G, Malinski T, Linz W. Decreased nitric oxide availability in normotensive and hypertensive rats with failing hearts after myocardial infarction. Hypertension. 2001;38: 1367–1371. doi:10.1161/hy1101.096115

15. Chen J, Petrov A, Yaniz-Galende E, Liang L, de Haas HJ, Narula J, et al. The impact of pressure overload on coronary vascular changes following myocardial infarction in rats. Am J Physiol Heart Circ Physiol. 2013;304: H719-28. doi:10.1152/ajpheart.00793.2012

16. Pfeffer M, Pfeffer J, Steinberg C, Finn P. Survival after an experimental myocardial infarction: beneficial effects of long-term therapy with captopril. Circulation. 1985;72: 406–412. doi:10.1161/01.CIR.72.2.406

17. Pfeffer M a, Pfeffer JM, Fishbein MC, Fletcher PJ, Spadaro J, Kloner R a, et al. Myocardial infarct size and ventricular function in rats. Circ Res. 1979;44: 503–512. doi:10.1161/01.RES.44.4.503

18. Patten RD, Hall-Porter MR. Small Animal Models of Heart Failure: Development of Novel Therapies, Past and Present. Circ Hear Fail. 2009;2: 138–144. doi:10.1161/CIRCHEARTFAILURE.108.839761

19. Doggrell S, Brown L. Rat models of hypertension, cardiac hypertrophy and failure. Cardiovasc Res. 1998;39: 89–105. doi:10.1016/s0008-6363(98)00076-5

20. The RALES investigators. Effectiveness of spironolactone added to an angiotensin-converting enzyme inhibitor and a loop diuretic for severe chronic congestive heart failure (the Randomized Aldactone Evaluation Study [RALES]). Am J Cardiol. 1996;78: 902–907. doi:10.1016/S0002-9149(96)00465-1

21. Leader CJ, Moharram M, Coffey S, Sammut IA, Wilkins GW, Walker RJ. Myocardial global longitudinal strain: An early indicator of cardiac interstitial fibrosis modified by spironolactone, in a unique hypertensive rat model. PLoS One. 2019;14: e0220837. doi:10.1371/journal.pone.0220837

22. Lichtenauer M, Schreiber C, Jung C, Beer L, Mangold A, Gyöngyösi M, et al. Myocardial infarct size measurement using geometric angle calculation. Eur J Clin Invest. 2014;44: 160–167. doi:10.1111/eci.12202

23. Ronco C, Ronco F. Cardio-renal syndromes: A systematic approach for consensus definition and classification. Heart Fail Rev. 2012;17: 151–160. doi:10.1007/s10741-010-9224-0

24. Charan J, Kantharia N. How to calculate sample size in animal studies? J Pharmacol Pharmacother. 2013;4: 303. doi:10.4103/0976-500X.119726

25. Nagata K. Mineralocorticoid antagonism and cardiac hypertrophy. Curr Hypertens Rep. 2008;10: 216–221. doi:10.1007/s11906-008-0041-y 

26. Ashek A, Menzies RI, Mullins LJ, Bellamy COC, Harmar AJ, Kenyon CJ, et al. Activation of thiazide-sensitive co-transport by angiotensin II in the cyp1a1-Ren2 hypertensive rat. PLoS One. 2012;7: e36311. doi:10.1371/journal.pone.0036311

---

## [Decision Letter · Decision Letter 1]

12 Nov 2021

The effect of spironolactone on cardiac and renal fibrosis following myocardial infarction in established hypertension in the transgenic Cyp1a1Ren2 rat.

PONE-D-21-28323R1

Dear Dr. Walker,

We’re pleased to inform you that your manuscript has been judged scientifically suitable for publication and will be formally accepted for publication once it meets all outstanding technical requirements.

Kind regards,

Vincenzo Lionetti, M.D., PhD

Academic Editor

PLOS ONE

Additional Editor Comments (optional):

Reviewers' comments:

Reviewer's Responses to Questions

**Comments to the Author**

1. If the authors have adequately addressed your comments raised in a previous round of review and you feel that this manuscript is now acceptable for publication, you may indicate that here to bypass the “Comments to the Author” section, enter your conflict of interest statement in the “Confidential to Editor” section, and submit your "Accept" recommendation.

Reviewer #1: All comments have been addressed

Reviewer #2: All comments have been addressed

2. Is the manuscript technically sound, and do the data support the conclusions?

Reviewer #1: Yes

Reviewer #2: Yes

3. Has the statistical analysis been performed appropriately and rigorously? 

Reviewer #1: Yes

Reviewer #2: Yes

4. Have the authors made all data underlying the findings in their manuscript fully available?

Reviewer #1: Yes

Reviewer #2: Yes

5. Is the manuscript presented in an intelligible fashion and written in standard English?

Reviewer #1: Yes

Reviewer #2: Yes

6. Review Comments to the Author

Reviewer #1: Authors have adequately addressed the questions raised in review. Some aspects remain ambiguous and data quantification (ie images) remains an issue. But this is the field as it sands as authors indicated.

Reviewer #2: (No Response)

7. PLOS authors have the option to publish the peer review history of their article (what does this mean?). If published, this will include your full peer review and any attached files.

Reviewer #1: No

Reviewer #2: No

---

## [Editor Report · Acceptance letter]

16 Nov 2021

PONE-D-21-28323R1 

The effect of spironolactone on cardiac and renal fibrosis following myocardial infarction in established hypertension in the transgenic Cyp1a1Ren2 rat. 

Dear Dr. Walker:

I'm pleased to inform you that your manuscript has been deemed suitable for publication in PLOS ONE. Congratulations! Your manuscript is now with our production department. 

Kind regards, 

on behalf of

Prof. Vincenzo Lionetti 

Academic Editor

PLOS ONE